# SEMI-SUPERVISED LEARNING WITH NORMALIZING FLOWS

## ABSTRACT

We propose Flow Gaussian Mixture Model (FlowGMM), a general-purpose method for semi-supervised learning based on a simple and principled probabilistic framework. We approximate the joint distribution of the labeled and unlabeled data with a flexible mixture model implemented as a Gaussian mixture transformed by a normalizing flow. We train the model by maximizing the exact joint likelihood of the labeled and unlabeled data. We evaluate FlowGMM on a wide range of semi-supervised classification problems across different data types: AG-News and Yahoo Answers text data, MNIST, SVHN and CIFAR-10 image classification problems as well as tabular UCI datasets. FlowGMM achieves promising results on image classification problems and outperforms the competing methods on other types of data. FlowGMM learns an interpretable latent representation space and allows hyper-parameter free feature visualization at real time rates. Finally, we show that FlowGMM can be calibrated to produce meaningful uncertainty estimates for its predictions.

## 1 INTRODUCTION

In many domains unlabeled data is plentiful, while labeled data may be scarce. Semi-supervised learning framework leverages both labeled and unlabeled data reducing the need for expensive manual annotation. Recently, consistency-based methods have shown outstanding performance in semi-supervised image classification (Laine & Aila, 2016; Miyato et al., 2018; Tarvainen & Valpola, 2017; Athiwaratkun et al., 2019; Verma et al., 2019; Berthelot et al., 2019) and are currently state-of-the-art on challenging datasets like CIFAR-10, CIFAR-100, and ImageNet. However, these methods have not seen much application on domains other than images, where a suitable set of data perturbations to which the classifier should be invariant is not known a priori.

In this paper, we employ a simple and principled probabilistic framework for semi-supervised learning. We introduce a mixture distribution for modeling the data, where different components correspond to different classes. We can maximize the joint likelihood of unlabeled (using the mixture) and labeled (using individual mixture components) data. At test time, we classify the input to belong to the class corresponding to mixture component with the highest likelihood.

In order to apply this framework to complex data, we need to choose a sufficiently flexible family of distributions in the mixture, preserving the ability to compute exact likelihoods. We propose Flow Gaussian Mixture Model (FlowGMM), a mixture model based on normalizing flows (Dinh et al., 2014). Each component of the mixture is modeled as a trainable invertible transformation (normalizing flow) of the corresponding component of a Gaussian mixture in the latent space. The transformation is shared among all the mixture components. Due to invertibility, we can compute exact likelihood of the data using the change of variable formula.

We illustrate FlowGMM on a toy problem in Figure 1. We are solving a binary semi-supervised classification problem on the dataset shown in panel (a): the labeled data is shown with triangles colored according to their class, and unlabeled data is shown with blue circles. We introduce a Gaussian mixture with two components corresponding to each of the classes, shown in panel (c) in the latent space $\mathcal{Z}$ and an invertible transformation $f$. The transformation $f$ is then trained to map the data distribution in the data space $\mathcal{X}$ to the latent Gaussian mixture in the $\mathcal{Z}$ space, mapping the labeled data to the corresponding mixture component. We visualize the learned transformation in panel (b), showing the positions of the images $f(x)$ for all of the training data points. The inverse

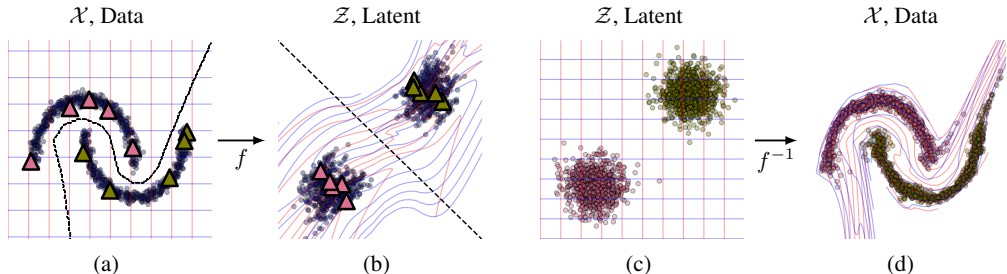

$\mathcal{X}$, Data     $\mathcal{Z}$, Latent     $\mathcal{Z}$, Latent     $\mathcal{X}$, Data

(a)     (b)     (c)     (d)

Figure 1: Illustration of semi-supervised learning with FlowGMM on a binary classification problem. Colors represent the two classes or the corresponding Gaussian mixture components. Labeled data is shown with triangles, colored by the corresponding class label, and blue dots represent unlabeled data. **(a):** Data distribution and the classifier decision boundary. **(b):** The learned mapping of the data to the latent space. **(c):** Samples from the Gaussian mixture in the latent space. **(d):** Samples from the model in the data space.

$f^{-1}$ of this mapping serves as a class-conditional generative model, that we visualize in panel (d). To classify a data point $x$ in the input space we compute its image $f(x)$ in the latent space, and pick the class corresponding to the Gaussian that is closest to $f(x)$. We visualize the decision boundary of the learned classifier with a dashed line in panel (a).

Many of the previous attempts of using generative models for classification (Salimans et al., 2016; Nalisnick et al., 2019; Chen et al., 2019) have relied upon multitask learning, where a shared latent representation is learned for the generative model and the classifier. With the method of Chen et al. (2019), hybrid modeling is observed to *reduce* performance for both tasks in the supervised case. For GANs, Dai et al. (2017) have pointed out that classification performance and generative performance are in direct conflict: a perfect generator yields no benefit to classification performance. FlowGMM takes a different approach, where the generative model is used directly as a Bayes classifier, and in the limit of a perfect generative model the Bayes classifier achieves provably optimal misclassification rate (see e.g. Mohri et al., 2018).

FlowGMM naturally encodes the *clustering principle*: the decision boundary between classes must lie in the low-density region in the data space. Indeed, in the latent space the decision boundary between two classes coincides with the hyperplane perpendicular to the line segment connecting means of the corresponding mixture components and passing through the midpoint of this line segment (assuming the components are normal distributions with identity covariance matrices); in panel (b) of Figure 1 we show the decision boundary in the latent space with a dashed line. The density of the latent distribution near the decision boundary is low. As the flow is trained to represent data as a transformation of this latent distribution, the density near the decision boundary should also be low. In panel (a) of Figure 1 the decision boundary indeed lies in the low-density region.

The contributions of this work are as follows:

- We propose FlowGMM, a new probabilistic classification model based on normalizing flows, that can be naturally applied to semi-supervised learning. We evaluate FlowGMM on a range of semi-supervised classification benchmarks including text, tabular and image data, and it works well. FlowGMM outperforms alternative approaches on non-image data and shows promise for image data. We propose modified consistency regularization for FlowGMM and empirically demonstrate that it substantially improves performance of the method on image classification problems.

- We conduct a thorough empirical analysis of FlowGMM for supervised and semi-supervised classification. One of the important features of FlowGMM is its interpretability. To demonstrate it, we visualize the learned latent space representations for the proposed semi-supervised model and show that interpolations between data points from different classes pass through low-density regions. We show how our classification model can be used for optimization free feature visualization. We also study the predictive uncertainties produced by the method and show that they can be calibrated by scaling the variances of mixture components.

## 2 BACKGROUND: NORMALIZING FLOWS

The normalizing flow (Dinh et al., 2016) is an unsupervised model for density estimation defined as an invertible mapping $f : \mathcal{X} \to \mathcal{Z}$ from the data space $\mathcal{X}$ to the latent space $\mathcal{Z}$. We can model the data distribution as a transformation $f^{-1} : \mathcal{Z} \to \mathcal{X}$ applied to a random variable from the latent distribution $z \sim p_{\mathcal{Z}}$ often chosen to be Gaussian. The density of the transformed random variable $x = f^{-1}(z)$ is given by the change of variables formula

$$p_{\mathcal{X}}(x) = p_{\mathcal{Z}}(f(x)) \cdot \left| \det\left(\frac{\partial f}{\partial x}\right) \right|. \tag{1}$$

The mapping $f$ is implemented as a sequence of invertible functions, parametrized by a neural network with architecture that is designed to ensure invertibility and efficient computation of log-determinants, and a set of parameters $\theta$ that can be optimized. The model can be trained by maximizing the likelihood (equation 1) of the training data with respect to the parameters $\theta$.

## 3 FLOW GAUSSIAN MIXTURE MODEL (FLOWGMM)

In FlowGMM, we introduce a discrete latent variable $y$ for the class label, $y \in \{1 \dots \mathcal{C}\}$. Our latent space distribution, conditioned on a given label $k$, is Gaussian with mean $\mu_k$ and covariance $\Sigma_k$:

$$p_{\mathcal{Z}}(z|y = k) = \mathcal{N}(z|\mu_k, \Sigma_k). \tag{2}$$

The marginal distribution of $z$ is then a Gaussian mixture. When the classes are balanced, this distribution is

$$p_{\mathcal{Z}}(z) = \frac{1}{\mathcal{C}} \sum_{k=1}^{\mathcal{C}} \mathcal{N}(z|\mu_k, \Sigma_k). \tag{3}$$

Thus, the likelihood for labeled data is

$$p_{\mathcal{X}}(x|y = k) = \mathcal{N}\left(f(x)|\mu_k, \Sigma_k\right) \cdot \left| \det\left(\frac{\partial f}{\partial x}\right) \right|,$$

and the likelihood for data with unknown label is $p_{\mathcal{X}}(x) = \sum_k p_{\mathcal{X}}(x|y = k)p(y = k)$. If we have access to both a labeled dataset $\mathcal{D}_\ell$ and an unlabeled dataset $\mathcal{D}_u$, then we can train our model in a semi-supervised way to maximize the joint likelihood of the labeled and unlabeled data

$$p_{\mathcal{X}}(\mathcal{D}_\ell, \mathcal{D}_u) = \prod_{(x_i, y_i) \in \mathcal{D}_\ell} p_{\mathcal{X}}(x_i, y_i) \prod_{x_j \in \mathcal{D}_u} p_{\mathcal{X}}(x_j), \tag{4}$$

over the parameters $\theta$ of the bijective function $f$, which learns a density model with a Bayes classifier: given a test point $x$, the model predictive distribution is given by

$$p_{\mathcal{X}}(y|x) = p_{\mathcal{X}}(x|y)p(y)/p(x) = \frac{\mathcal{N}\left(f(x)|\mu_y, \Sigma_y\right)}{\sum_{k=1}^{\mathcal{C}} \mathcal{N}(f(x)|\mu_k, \Sigma_k)}. \tag{5}$$

We can then make predictions for a test point $x$ with the Bayes decision rule

$$y = \arg \max_{i \in \{1, \dots, \mathcal{C}\}} p_{\mathcal{X}}(y = i|x).$$

Alternatively to direct likelihood maximization, we can adapt Expectation Maximization algorithm for model training (discussed in Appendix A).

### 3.1 CONSISTENCY REGULARIZATION

Most of the existing state-of-the-art approaches to semi-supervised learning on image data are based on consistency regularization (Laine & Aila, 2016; Miyato et al., 2018; Tarvainen & Valpola, 2017; Athiwaratkun et al., 2019; Verma et al., 2019). These methods penalize changes in network predictions with respect to input perturbations, such as random translations and horizontal flips, with an additional loss term that can be computed on unlabeled data,

$$\ell_{cons}(x) = \|g(x') - g(x'')\|^2, \tag{6}$$

where $x', x''$ are random perturbations of $x$, and $g$ is the vector of probabilities over the classes.

Motivated by these methods, we introduce a simple consistency regularization term for FlowGMM. Let $y''$ be the label predicted on image $x''$ by FlowGMM according to equation 5. We then define the consistency loss term as the negative log likelihood of the input $x'$ given the label $y''$:

$$L_{\text{cons}}(x', x'') = -\log p(x'|y'') = -\log \mathcal{N}(f(x')|\mu_{y''}, \Sigma_{y''}) - \log \left| \det \left( \frac{\partial f}{\partial x'} \right) \right|. \tag{7}$$

This loss term encourages the model to map small perturbations of the same unlabeled inputs to the same components of the Gaussian mixture distribution in the latent space. Unlike the standard consistency loss of equation 6, the proposed loss in equation 7 takes values on the same scale as the data log likelihood (equation 4), and empirically we found it to perform better. We refer to FlowGMM with the consistency term as FlowGMM-cons. The final loss for FlowGMM-cons is then the weighted sum of the consistency loss (equation 7) and the negative log likelihood of both labeled and unlabeled data (equation 4).

## 4 RELATED WORK

Generative models have been used extensively in semi-supervised learning. In the work of Kingma et al. (2014), it was shown how the likelihood model of Variational Autoencoder (Kingma & Welling, 2013) could be used for semi-supervised image classification on datasets like MNIST and SVHN. Xu et al. (2017) later extended this framework to semi-supervised text classification. Generative Adversarial Networks (GANs) have been employed for semi-supervised learning through multitask objective where the model learns to simultaneously discriminate generated images from real (labeled and unlabeled) images and classify labeled data.

Along with GANs and VAEs, Normalizing Flows (NF) (Dinh et al., 2014) present another major class of deep generative models. Unlike GANs and VAEs, normalizing flows can be trained using exact likelihood. NFs admit controllable latent representations and can be sampled efficiently unlike auto-regressive models (Papamakarios et al., 2017; Oord et al., 2016). Recent work (Dinh et al., 2016; Kingma & Dhariwal, 2018; Behrmann et al., 2018) demonstrated that normalizing flows can produce high-fidelity samples for natural image datasets. Some normalizing flow papers (such as RealNVP (Dinh et al., 2016)) have used class-conditional sampling, where the transformation is conditioned on the class label. To do so, they pass the class label as an input to coupling layers, conditioning the output of the flow on the class.

Deep Invertible Generalized Linear Model (DIGLM, Nalisnick et al., 2019), most closely related to our work, trains a classifier on the latent representation of a normalizing flow to perform supervised or semi-supervised image classification. Our approach is principally different, as we use a mixture of Gaussians in the latent space $\mathcal{Z}$ and perform classification based on class-conditional likelihoods (see equation 5), rather than training a separate classifier. One of the key advantages of our approach is the explicit encoding of clustering principle in the method and a more natural probabilistic interpretation.

## 5 EXPERIMENTS

In all experiments, we use the RealNVP normalizing flow architecture. Throughout training, Gaussian mixture parameters are fixed: the means are initialized randomly from the standard normal distribution and the covariances are set to $I$. See Appendix B for further discussion on GMM initialization and training.

### 5.1 SYNTHETIC DATA

We first apply FlowGMM to a range of two-dimensional synthetic datasets, in order to get a better visual intuition for the method. We use RealNVP architecture with 5 coupling layers, defined by fully-connected shift and scale networks, each with 1 hidden layer of size 512. In addition to the semi-supervised setting, we also trained the method only using the labeled data. In Figure 2 we visualize the decision boundaries of the classifier corresponding to FlowGMM for both of these

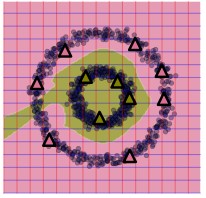 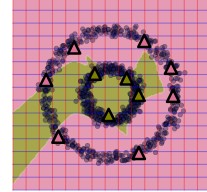 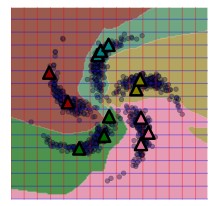 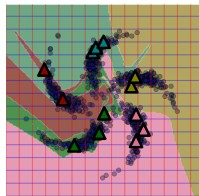

| (a) Labeled + Unlabeled | (b) Labeled Only | (c) Labeled + Unlabeled | (d) Labeled Only |

Figure 2: Illustration of FlowGMM performance on synthetic datasets. Labeled data is shown with colored triangles, and unlabeled data is shown with blue circles. Colors represent different classes. We compare the classifier decision boundaries when only using labeled data (panels b, d) and when using both labeled and unlabeled data (panels a, c) on two circles (panels a, b) and pinwheel (panels c, d) datasets. FlowGMM leverages unlabeled data to push the decision boundary to low-density regions of the space.

Table 1: Accuracy on BERT embedded text classification datasets and UCI datasets with a small number of labeled examples. The kNN baseline, logistic regression, and the 3-Layer NN + Dropout were trained on the labeled data only. Numbers reported for each method are the best of 3 runs (ranked by performance on the validation set). $n_l$ and $n_u$ are the number of labeled and unlabeled data points.

| | Dataset ($n_l$ / $n_u$, classes) | | | |
|---|---|---|---|---|
| Method | AG-News (200 / 200k, 4) | Yahoo Answers (800 / 50k, 10) | Hepmass (20 / 140k, 2) | Miniboone (20 / 65k, 2) |
| kNN | 51.3 | 28.4 | 84.6 | 77.7 |
| Logistic Regression | 78.9 | 54.9 | 84.9 | 75.9 |
| 3-Layer NN + Dropout | 78.1 | 55.6 | 84.4 | 77.3 |
| RBF Label Spreading | 54.6 | 30.4 | 87.1 | 78.8 |
| kNN Label Spreading | 56.7 | 25.6 | 87.2 | 78.1 |
| Π-model | 80.6 | 56.6 | 87.9 | 78.3 |
| FlowGMM | **84.8** | **57.4** | **88.8** | **80.6** |

settings on the *two circles* and *pinwheel* datasets. On both datasets FlowGMM is able to leverage the unlabeled data to push the decision boundary to a low-density region, as expected. On the two circles dataset the method is unable to fit the data perfectly, as it is impossible to represent this dataset as an invertible continuous mapping of two Gaussians, because they are topologically distinct. FlowGMM still produces a reasonable decision boundary and improves over the case when only labeled data is available. We provide additional visualizations in Appendix C, Figure 4.

## 5.2 TEXT AND TABULAR DATA

We believe that FlowGMM can be especially useful for semi-supervised learning on tabular data. Consistency-based semi-supervised methods have mostly been developed for image classification, where the predictions of the method are regularized to be invariant to random flips and translations of the image. On tabular data, where the structure is less prominent, finding suitable transformations to apply for consistency-based methods is not-trivial. Similarly, approaches based on GANs have mostly been developed for images. We evaluate FlowGMM on the Hepmass and Miniboone UCI classification datasets (previously used in Papamakarios et al. (2017) for density estimation).

Along with standard tabular UCI datasets, we also consider text classification on AG-News and Yahoo Answers datasets. Using the recent advances in transfer learning for NLP, we construct embeddings for input texts using the BERT transformer model (Devlin et al., 2018) trained on a corpus of Wikipedia articles, and then train FlowGMM and other baselines on the embeddings.

We compare FlowGMM to the graph based label spreading method from Zhou et al. (2004), a Π-Model (Laine & Aila, 2016) that uses dropout perturbations, as well as supervised logistic regression, k-nearest neighbors, and neural network that were trained on the labeled data only. We report

Table 2: Accuracy of the FlowGMM, VAE model (M1+M2 VAE, Kingma et al., 2014), DIGLM (Nalisnick et al., 2019) in supervised and semi-supervised settings on MNIST, SVHN, and CIFAR-10. FlowGMM Sup (*All* labels) as well as DIGLM Sup (*All* labels) were trained on full train datasets with all labels to demonstrate general capacity of these models. FlowGMM Sup ($n_l$ labels) was trained on $n_l$ labeled examples (and no unlabeled data). For reference, at the bottom we list the performance of the Π-Model (Laine & Aila, 2016) and BadGAN (Dai et al., 2017) as representative consistency-based and GAN-based state-of-the-art methods. Both of these methods use non-invertible architectures with substantially higher base performance and, thus, are not directly comparable.

| Method | Dataset ($n_l$ / $n_u$) | | |
| --- | --- | --- | --- |
| | MNIST $(1k/59k)$ | SVHN $(1k/72k)$ | CIFAR-10 $(4k/46k)$ |
| DIGLM Sup (*All* labels) | 99.27 | 95.74 | - |
| FlowGMM Sup (*All* labels) | 99.63 | 95.81 | 88.44 |
| M1+M2 VAE SSL | 97.60 | 63.98 | - |
| DIGLM SSL | **99.0** | - | - |
| FlowGMM Sup ($n_l$ labels) | 97.36 | 78.26 | 73.13 |
| FlowGMM | 98.94 | 82.42 | 78.24 |
| FlowGMM-cons | **99.0** | **86.44** | **80.9** |
| BadGAN | - | 95.75 | 85.59 |
| Π-Model | - | 94.57 | 87.64 |

the results in Table 5.1, where FlowGMM beats the competing semi-supervised learning methods on each of the considered datasets. Implementation details for FlowGMM, the baselines, and the dataset preprocessing details are listed in Appendix D.

## 5.3 IMAGE CLASSIFICATION

We next evaluate the proposed method on semi-supervised image classification benchmarks on CIFAR-10, MNIST and SVHN datasets. For all the datasets, we use RealNVP (Dinh et al., 2016) architecture. Exact implementation details are listed in the appendix E. The supervised model is trained using the same loss (equation 4), where all the data points are labeled ($n_u = 0$).

We present the results for FlowGMM and FlowGMM-cons in Table 2. We also report results from DIGLM (Nalisnick et al., 2019) (which only report semi-supervised performance on MNIST and supervised performance on MNIST and SVHN) and the M1+M2 VAE model (Kingma et al., 2014). FlowGMM outperforms M1+M2 model and performs better or on par with DIGLM. Furthermore, FlowGMM-cons improves over FlowGMM on all three datasets, suggesting that consistency regularization is very beneficial for the proposed model when useful perturbations are available.

Following Oliver et al. (2018), we evaluate FlowGMM-cons varying the number of labeled data points. Specifically, we follow the setup of Kingma et al. (2014) and train FlowGMM-cons on MNIST with 100, 600, 1000 and 3000 labeled data points. We present the results in Table 3. FlowGMM-cons outperforms the M1+M2 model of Kingma et al. (2014) in all the considered settings.

We note that the results presented in this section are not directly comparable with the state-of-the-art methods using GANs or consistency regularization (see e.g. Laine & Aila, 2016; Dai et al., 2017; Athiwaratkun et al., 2019; Berthelot et al., 2019), as the architecture we employ is much less powerful for classification than the ConvNet and ResNet architectures that have been designed for classification without the constraint of invertibility. We believe that invertible architectures with better inductive biases for classification (possibly like iResNet (Behrmann et al., 2018)) may help bridge this gap. The space of images is challenging to model and while there has been substantial progress using normalizing flows, there is still much ground to cover.

Table 3: Semi-supervised classification accuracy for FlowGMM-cons and VAE M1 + M2 model (Kingma et al., 2014) on MNIST for different number of labeled data points $n_l$.

| Method | $n_l = 100$ | $n_l = 600$ | $n_l = 1000$ | $n_l = 3000$ |
|---|---|---|---|---|
| M1+M2 VAE SSL ($n_l$ labels) | 96.67 | $97.41 \pm 0.05$ | $97.60 \pm 0.02$ | $97.82 \pm 0.04$ |
| FlowGMM-cons ($n_l$ labels) | 98.2 | 98.7 | 99 | 99.2 |

Table 4: Negative log-likelihood and Expected Calibration Error for supervised FlowGMM trained on MNIST (1k train, 1k validation, 10k test) and CIFAR-10 (50k train, 1k validation, 9k test). FlowGMM-temp stands for tempered FlowGMM where a single scalar parameter $\sigma^2$ was learned on a validation set for variances in all components.

| | MNIST (test acc 97.3%) | | CIFAR-10 (test acc 89.3%) | |
|---|---|---|---|---|
| | FlowGMM | FlowGMM-temp | FlowGMM | FlowGMM-temp |
| NLL $\downarrow$ | 0.295 | 0.094 | 2.98 | 0.444 |
| ECE $\downarrow$ | 0.024 | 0.004 | 0.108 | 0.038 |

## 6 MODEL ANALYSIS

### 6.1 UNCERTAINTY AND CALIBRATION

In many machine learning applications, it is crucial to understand how confident a model is in its predictions. In classification problems, well-calibrated models are expected to output meaningful probabilities of belonging to a particular class. In Guo et al. (2017), it was observed that modern deep learning models are highly overconfident, however, simple temperature scaling can substantially improve model's calibration. In this section, we analyze the predictive uncertainties produced by the FlowGMM. In Appendix Section F, we additionally study the ability of FlowGMM to detect out-of-domain data.

When using FlowGMM for classification, the class predictive probabilities are $p(y = k|x) = \mathcal{N}(x|\mu_k, \Sigma_k)/\sum_m \mathcal{N}(x|\mu_m, \Sigma_m)$. Since we initialize Gaussian mixture means randomly from the standard normal distribution and do not train them along with the flow parameters (see section B), FlowGMM predictions become inherently overconfident due to the curse of dimensionality. Indeed, consider two Gaussians with means sampled independently from the standard normal $\mu_1, \mu_2 \sim \mathcal{N}(0, I)$ in $D$-dimensional space. If $s_1 \sim \mathcal{N}(\mu_1, I)$ is a sample from the first Gaussian, then its expected squared distances to both mixture means are $\mathbb{E}\left[\|s_1 - \mu_1\|^2\right] = D$ and $\mathbb{E}\left[\|s_1 - \mu_2\|^2\right] = 3D$ (for a detailed derivation see Appendix Section G.). In high dimensional spaces, such logits would lead to hard label assignment in FlowGMM ($p(y|x) = 1$ for exactly one class). In fact, in the experiments we observe that FlowGMM is overconfident and performs hard label assignment: predicted class probabilities are all close to either 1 or 0.

We address this problem by learning a single scalar parameter $\sigma^2$ for all components in the Gaussian mixture (the component $k$ will be $\mathcal{N}(\mu_k, \sigma^2 I)$) by minimizing the negative log likelihood on a validation set. This way we can re-calibrate the variance of the latent GMM in a natural way. This procedure is also equivalent to applying temperature scaling (Guo et al., 2017) to logits $\log \mathcal{N}(x|\mu_k, \Sigma_k)$. We test FlowGMM calibration on MNIST and CIFAR datasets in the supervised setting. On MNIST we restricted the training set size to 1000 objects, since on the full dataset the model makes too few mistakes which makes evaluating calibration harder. In Table 4, we report negative log likelihood and expected calibration error (ECE, see Guo et al. (2017) for a description of this metric). We can see that re-calibrating variances of the Gaussians in the mixture significantly improves both metrics and mitigates overconfidence. The effectiveness of this simple rescaling procedure suggests that the latent space distances learned by the flow model are correlated with the probabilities of belonging to a particular class: the closer a datapoint is to the mean of a Gaussian in the latent space, the more likely it belongs to the corresponding class.

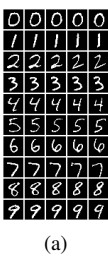 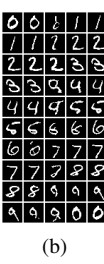 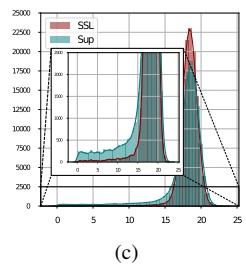 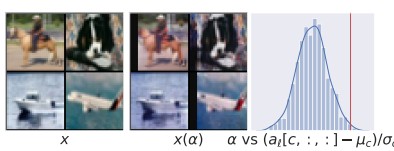

(a)   (b)   (c)   (d)

Figure 3: Visualizations of the latent space representations learned by supervised FlowGMM on MNIST. **(a)**: Latent space interpolations between test images from the same class and **(b)**: from different classes. Observe that interpolations between objects from different classes pass through low-density regions. **(c)**: Histogram of distances from unlabeled data to the decision boundary for FlowGMM-cons trained on $1k$ labeled and $59k$ unlabeled data and FlowGMM Sup trained on $1k$ labeled data only. FlowGMM-cons is able to push the decision boundary away from the data distribution using unlabeled data. **(d)**: Feature visualization for CIFAR10: four test reconstructions are shown as an intermediate feature is perturbed. The value of the perturbation $\alpha$ is shown in red vs the distribution of the channel activations. Observe that the channel visualized activates on zeroed out pixels to the left of the image mimicking the random translations applied to the training data.

## 6.2 LEARNED LATENT REPRESENTATIONS

We next analyze the latent representation space learned by FlowGMM. We examine latent interpolations between members of the same class in Figure 3 (a) and between different classes in Figure 3 (b) for our MNIST FlowGMM-cons model trained with $n_\ell = 1k$ labels. As expected, inter-class interpolations pass through regions of low-density, leading to low quality samples but intra-class interpolations do not. These observations suggest that, as expected, the model learns to put the decision boundary in the low-density region of the data space.

In Appendix section H, we present images corresponding to the means of the Gaussian mixture and class-conditional samples from FlowGMM.

**Distance to Decision Boundary** To explicitly test this conclusion, we compute the distribution of distances from unlabeled data to the decision boundary for FlowGMM-cons and FlowGMM Sup trained on labeled data only. In order to compute this distance exactly for an image $x$, we find the two closest means $\mu'$, $\mu''$ to the corresponding latent variable $z = f(x)$, and evaluate the expression $d(x) = \frac{\left| \|\mu' - f(x)\|^2 - \|\mu'' - f(x)\|^2 \right|}{2\|\mu' - \mu''\|}$. We visualize the distributions of the distances for the supervised and semi-supervised method in Figure 3 (c). While most of the unlabeled data are far from the decision boundary for both methods, the supervised method puts a substantially larger fraction of data close to the decision boundary. For example, the distance to the decision boundary is smaller than 5 for 1089 unlabeled data points with supervised model, but only 143 data points with FlowGMM-cons. This increased separation suggests that FlowGMM-cons indeed pushes the decision boundary away from the data distribution in agreement with the clustering principle.

## 6.3 FEATURE VISUALIZATION

Feature visualization has become an important tool for increasing the interpretability of neural networks. The majority of methods rely on maximizing the activations of a given neuron, channel, or layer over a parametrization of an input image with different kinds of image regularization (Szegedy et al., 2013; Olah et al., 2017; Mahendran & Vedaldi, 2015). These methods, while effective, require optimization and regularization hyper-parameters and iterative optimization too costly for real time interactive exploration.

Since our classification model uses a flow which is a sequence of invertible transformations $f(x) = f_{:L}(x) := f_L \circ f_{L-1} \circ \ldots f_1(x)$, intermediate activations can be inverted directly. This means that we can combine the methods of feature inversion and feature maximization directly by feeding in a set of input images, modifying intermediate activations arbitrarily, and inverting the representation.

Given a set of activations in the $\ell^{th}$ layer $a_\ell[c, i, j] = f_{:\ell}(x)_{cij}$ with channels $c$ and spatial extent $i, j$, we may perturb a single neuron with

$$x(\alpha) = f_{:\ell}^{-1}(f_{:\ell}(x) + \alpha\sigma_c\delta_c), \tag{8}$$

where $\delta_c$ is a one hot vector at channel $c$; and $\sigma_c$ is the standard deviation of the activations in channel $c$ over the the training set and spatial locations. This can be performed at real time rates to explore the activation parametrized by $\alpha$ and the location $cij$ without any optimization or hyper-parameters. The feature visualization of intermediate layers on CIFAR10 test images are shown in panel (d) of Figure 3. The given channel being visualized appears to activate on the zeroed pixels from random translations as well as the green channel, giving us insight into the workings of the model.

## 7 DISCUSSION

We proposed FlowGMM, a natural and interpretable model for semi-supervised learning with normalizing flows. FlowGMM outperforms graph-based and consistency-based baselines on tabular data including semi-supervised text classification with BERT embeddings. On image classification, FlowGMM is not yet competitive with the state-of-the-art approaches (Athiwaratkun et al., 2019; Verma et al., 2019), but we expect that the results could be greatly improved by using a better flow architecture that has stronger inductive biases for classification.

FlowGMM is greatly extensible and can be easily adapted to different settings. For example, consider an extreme case of semi-supervised learning where some classes may be completely unlabeled, and the model has to identify those extra classes and cluster them separately. An effective model that can operate in this setting could, for example, be used for automated scientific discovery: when applied to classify certain experimental data, it can identify new separate classes that were not considered by experts. FlowGMM can be applied in this scenario: instead of the standard Gaussian mixture model (GMM) in the latent space we can use the Chinese Restaurant Process GMM (CRP-GMM), which can infer the number of mixture components from data automatically (Rasmussen, 2000).

Further, we can easily adapt FlowGMM to few-shot learning setting where the model has to generalize to classes not seen at train time by only using a few examples of that new class. In particular, we can fix the parameters of the flow model and add a new Gaussian to the mixture whenever we add a new class. We can estimate the parameters of this Gaussian from the few available examples.

We view interpretability as another strong advantage of FlowGMM. The access to latent space representations and feature visualization technique discussed in Section 6 as well as the ability to sample from the model can be used to get insights into the performance of the model in practical applications.

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

## A    EXPECTATION MAXIMIZATION

As an alternative to direct optimization of the likelihood (equation 4), we consider Expectation-Maximization algorithm (EM). EM is a popular approach for finding maximum likelihood estimates in mixture models. Suppose $X = \{x_i\}_{i=1}^n$ is the observed dataset, $T = \{t_i\}_{i=1}^n$ are corresponding unobserved latent variables (often denoting the component in mixture model) and $\theta$ is a vector of model parameters. EM algorithm consists of the two alternating steps: on E-step, we compute posterior probabilities of latent variables for each data point $q(t_i|x_i) = P(t_i|x_i, \theta)$; and on M-step, we fix $q$ and maximize the expected log likelihood of the data and latent variables with respect to $\theta$: $\mathbb{E}_q \log P(X, T|\theta) \to \max_\theta$. The algorithm can be easily adapted to the semi-supervised setting where a subset of data is labeled with $\{y_i^l\}_{i=1}^{n_l}$: then, on E-step we have hard assignment to the true mixture component $q(t_i|x_i) = I[t_i = y_i^l]$ for labeled data points.

EM algorithm is applicable in our setting which is fitting the transformed mixture of Gaussians. We can perform the exact E-step for unlabeled data in the model since

$$q(t|x) = \frac{p(x|t, \theta)}{p(x|\theta)} = \frac{\mathcal{N}(f(x)|\mu_t, \Sigma_t) \cdot \left|\det\left(\frac{\partial f}{\partial x}\right)\right|}{\sum_{k=1}^{\mathcal{C}} \mathcal{N}(f(x)|\mu_k, \Sigma_k) \cdot \left|\det\left(\frac{\partial f}{\partial x}\right)\right|} = \frac{\mathcal{N}(f(x)|\mu_t, \Sigma_t)}{\sum_{k=1}^{\mathcal{C}} \mathcal{N}(f(x)|\mu_k, \Sigma_k)}$$

which coincides with the E-step of EM algorithm on Gaussian mixture model. On M-step, the objective has the following form:

$$\sum_{i=1}^{n_l} \log \left[\mathcal{N}(f_\theta(x_i^l)|\mu_{y_i^l}, \Sigma_{y_i^l}) \left|\frac{\partial f_\theta}{\partial x_i^l}\right|\right] + \sum_{i=1}^{n_u} \mathbb{E}_{q(t_i|x_i^u, \theta)} \log \left[\mathcal{N}(f_\theta(x_i^u)|\mu_{t_i}, \Sigma_{t_i}) \left|\frac{\partial f_\theta}{\partial x_i^u}\right|\right].$$

Since the exact solution is not tractable due to complexity of the flow model, we perform a stochastic gradient step to optimize the expected log likelihood with respect to flow parameters $\theta$.

Note that unlike regular EM algorithm for mixture models, we have Gaussian mixture parameters $\{(\mu_k, \Sigma_k)\}_{k=1}^{\mathcal{C}}$ fixed in our experiments, and on M-step the update of $\theta$ induces the change of $z_i = f_\theta(x_i)$ latent space representations.

Using EM algorithm for optimization in the semi-supervised setting on MNIST dataset with 1000 labeled images, we obtain 98.97% accuracy which is comparable to the result for FlowGMM with regular SGD training. However, in our experiments, we observed that on E-step, hard label assignment happens for unlabeled points ($q(t|x) \approx 1$ for one of the classes) because of the high dimensionality of the problem (see section 6.1) which affects the M-step objective and hinders training.

## B    LATENT DISTRIBUTION MEAN AND COVARIANCE CHOICES

**Initialization**    In our experiments, we draw the mean vectors $\mu_i$ of Gaussian mixture model randomly from the standard normal distribution $\mu_i \sim \mathcal{N}(0, I)$, and set the covariance matrices to identity $\Sigma_i = I$ for all classes; we fixed GMM parameters throughout training. However, one could potentially benefit from data-dependent placing of means in the latent space. We experimented with different initialization methods, in particular, initializing means using the mean point of latent representations of labeled data in each class: $\mu_i = (1/n_l^i) \sum_{m=1}^{n_l^i} f(x_m^i)$ where $x_m^i$ represents labeled data points from class $i$ and $n_l^i$ is the total number of labeled points in that class. In addition, we can scale all means by a scalar value $\hat{\mu}_i = r\mu_i$ to increase or decrease distances between them. We observed that such initialization leads to much faster convergence of FlowGMM on semi-supervised classification on MNIST dataset, however, the final performance of the model was worse compared to the one with random mean placing. We hypothesize that it becomes easier for the flow model to warm up faster with data-dependent initialization because Gaussian means are closer to the initial latent representations, but afterwards the model gets stuck in a suboptimal solution.

**GMM training**    FlowGMM would become even more flexible and expressive if we could learn Gaussian mixture parameters in a principled way. In the current setup where means are sampled from the standard normal distribution, the distances between mixture components are about $\sqrt{2D}$ where $D$ is the dimensionality of the data (see Appendix G). Thus, classes are quite far apart from each

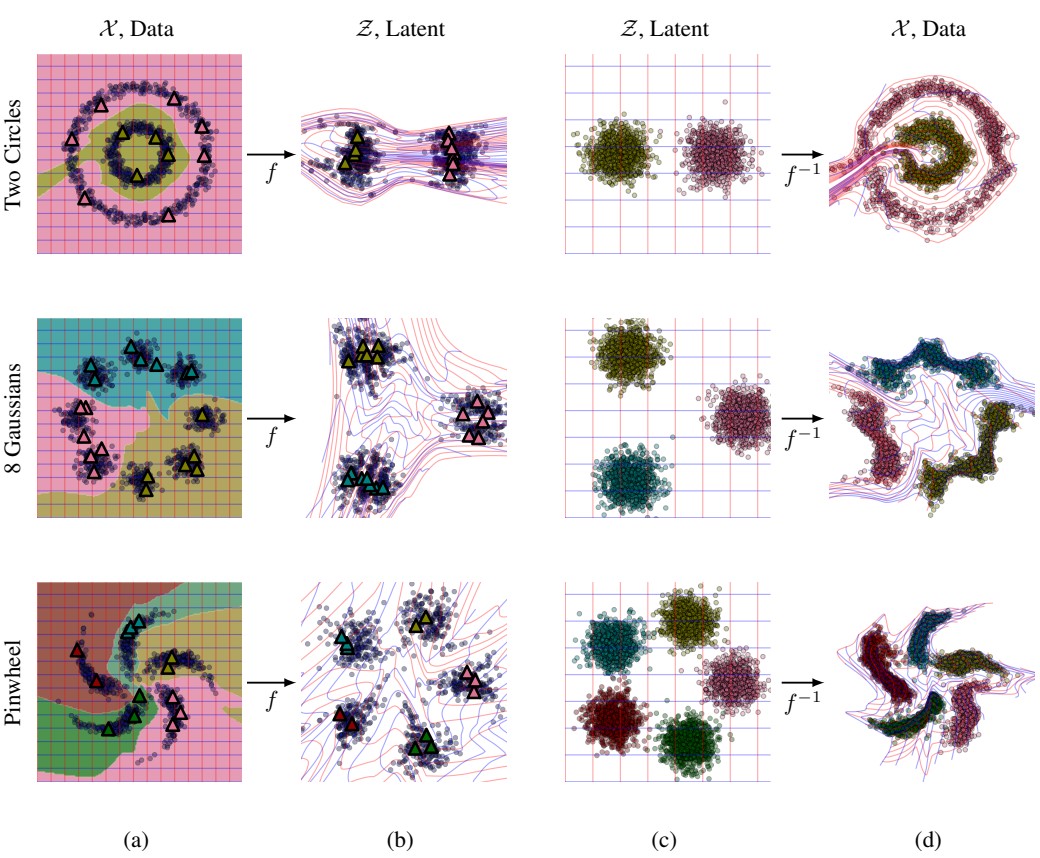

Figure 4: Illustration of FlowGMM on synthetic datasets: two circles (top row), eight Gaussians (middle row) and pinwheel (bottom row). **(a):** Data distribution and classification decision boundaries. Unlabeled data is shown with blue circles and labeled data is shown with colored triangles, where color represents the class. Background color visualizes the classification decision boundaries of FlowGMM. **(b):** Mapping of the data to the latent space. **(c):** Gaussian mixture in the latent space. **(d):** Samples from the learned generative model corresponding to different classes, as shown by their color.

other in the latent space, which, as observed in Section 6.1, leads to model miscalibration. Training GMM parameters can further increase interpretability of the learned latent space representations: we can imagine a scenario in which some of the classes are very similar or even intersecting, and it would be useful to represent it in the latent space. We could train GMM by directly optimizing likelihood (equation 4), or using expectation maximization (see Section A), either jointly with the flow parameters or iteratively switching between training flow parameters with the fixed GMM and training GMM with the fixed flow. In our initial experiments on semi-supervised classification on MNIST, training GMM jointly with the flow parameters did not improve performance or lead to substantial change of the latent representations. Further improvements require careful hyperparameter choice which we leave for future work.

## C    SYNTHETIC EXPERIMENTS

In Figure 4 we visualize the classification decision boundaries of FlowGMM as well as the learned mapping to the latent space and generated samples for three different synthetic datasets.

## D    TABULAR DATA PREPARATION AND HYPERPARAMETERS

The AG-News and Yahoo Answers were constructed by applying BERT embeddings to the text input, yielding a 768 dimensional vector for each data point. AG-News has 4 classes while Ya-

Table 5

| Method Learning Rate | AG-News | Yahoo Answers | Hepmass | Miniboone |
|---|---|---|---|---|
| 3-Layer NN + Dropout | 3e-4 | 3e-4 | 3e-4 | 3e-4 |
| Π-model | 1e-3 | 1e-4 | 3e-3 | 1e-4 |
| FlowGMM | 1e-4 | 1e-4 | 3e-3 | 3e-4 |
| kNN | $k = 4$ | $k = 18$ | $k = 9$ | $k = 3$ |

hoo Answers has 10. The UCI datasets Hepmass and Miniboone were constructed using the data preprocessing from Papamakarios et al. (2017), but with the inclusion of the removed background process class so that the two problems can be used for binary classification. We then subsample the fraction of background class examples so that the dataset is balanced. For each of the datasets, a separate validation set of size 5k was used to tune hyperparameters. All neural network models use the ADAM optimizer (Kingma & Ba, 2014).

**k-Nearest Neighbors**: We tested both using both L2 distance and L2 with inputs normalized to unit norm, ($\sin^2$ distance), and the latter performed the best. The value $k$ chosen in the method was found sweeping over $1 - 20$, and the optimal values for each of the datasets are shown in 5.

**3 Layer NN + Dropout**: The 3-Layer NN + Dropout baseline network has three fully connected hidden layers with inner dimension $k = 512$, ReLU nonlinearities, and dropout with $p = 0.5$. We use the learning rate $3e-4$ for training the supervised baseline across all datasets.

**Π-Model**: The Π-Model uses the same network architecture, and dropout for the perturbations. The additional consistency loss per unlabeled data point is computed as $L_{\text{Unlab}} = ||g(x'') - g(x')||^2$, where $g$ is are the output probabilities after the softmax layer of the neural network and the consistency weight $\lambda = 30$ which worked the best across the datasets. The model was trained for 50 epochs with labeled and unlabeled batch size $n_\ell$ for AG-News and Yahoo Answers, and labeled and unlabeled batch sizes $n_\ell$ and 2000 for Hepmass and Miniboone.

**Label Spreading**: We use the local and global consistency method from Zhou et al. (2004), $Y^* = (I - \alpha S)^{-1} Y$ where in our case $Y$ is the matrix of labels for the labeled, unlabeled, and test data but filled with zeros for unlabeled and test. $S = D^{-1/2} W D^{-1/2}$ computed from the affinity matrix $W_{ij} = \exp\left(-\gamma \sin^2(x_i, x_j)\right)$ where $\sin^2(x_i, x_j) := 1 - \frac{\langle x_i, x_j \rangle}{||x_i|| ||x_j||}$. This is equivalent to L2 distance on the inputs normalized to unit magnitude. Because the algorithm scales poorly with number of unlabeled points for dense affinity matrices, $O(n_u^3)$, we we subsampled the number of unlabeled data points to $10k$ and test data points to $5k$ for this graph method. However, we also evaluate the label spreading algorithm with a sparse kNN affinity matrix on using a larger subset $20k$ of unlabeled data. The two hyperparameters for label spreading ($\gamma/k$ and $\alpha$) were tuned by separate grid search for each of the datasets. In both cases, we use the inductive variant of the algorithm where the test data is not included in the unlabeled data.

**FlowGMM**: We train our FlowGMM model with a RealNVP normalizing flow, similar to the architectures used in Papamakarios et al. (2017). Specifically, the model uses 7 coupling layers, with 1 hidden layer each and 256 hidden units for the UCI datasets but 512 for text classification. UCI models were trained for 50 epochs of unlabeled data and the text datasets were trained for 30 epochs of unlabeled data. The labeled and unlabeled batch sizes are the same as in the Π-Model.

The tuned learning rates for each of the models that we used for these experiments are shown in table 5.

## E  IMAGE DATA PREPARATION AND HYPERPARAMETERS

We use the CIFAR-10 multi-scale architecture with 2 scales, each containing 3 coupling layers defined by 8 residual blocks with 64 feature maps. We use Adam optimizer (Kingma & Ba, 2014) with learning rate $10^{-3}$ for CIFAR-10 and SVHN and $10^{-4}$ for MNIST. We train the supervised model for 100 epochs, and semi-supervised models for 1000 passes through the labeled data for CIFAR-10 and SVHN and 3000 passes for MNIST. We use a batch size of 64 and sample 32 labeled and 32 unlabeled data points in each mini-batch. For the consistency loss term (equation 7), we linearly increase the weight from 0 to 1 for the first 100 epochs following Athiwaratkun et al. (2019).

For FlowGMM and FlowGMM-cons, we re-weight the loss on labeled data by $\lambda = 3$ (value tuned on validation (Kingma et al., 2014) on CIFAR-10), as otherwise, we observed that the method underfits the labeled data.

## F    OUT-OF-DOMAIN DATA DETECTION

Density models have held promise for being able to detect out-of-domain data, an especially important task for robust machine learning systems (Nalisnick et al., 2019). Recently, it has been shown that existing flow and autoregressive density models are not as apt at this task as previously thought, yielding high likelihood on images coming from other (simpler) distributions. The conclusion put forward is that datasets like SVHN are encompassed by, or have roughly the same mean but lower variance than, more complex datasets like CIFAR10 (Nalisnick et al., 2018). We examine this hypothesis in the context of our flow model which has a multi-modal latent space distribution unlike methods considered in Nalisnick et al. (2018).

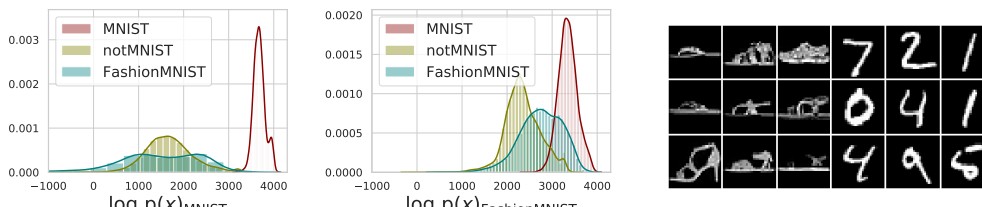

Figure 5: **Left:** Log likelihoods on in- and out-of-domain data for our model trained on MNIST. **Center:** Log likelihoods on in- and out-of-domain data for our model trained on FashionMNIST. **Right:** MNIST digits get mapped onto the sandal mode of the FashionMNIST model $75\%$ of the time, often being assigned higher likelihood than elements of the original sandal class. Representative elements are shown above.

Using a fully supervised model trained on MNIST, we evaluate the log likelihood for data points coming from the NotMNIST dataset, consisting of letters instead of digits, and the FashionMNIST dataset. We then train a supervised model on the more complex dataset FashionMNIST and evaluate on MNIST and NotMNIST. The distribution of the log likelihood $\log p_{\mathcal{X}}(\cdot) = \log p_{\mathcal{Z}}(f(\cdot)) + \log \left| \det \left( \frac{\partial f}{\partial x} \right) \right|$ on these datasets is shown in Figure 5. For the model trained on MNIST we see that the data from Fashion MNIST and NotMNIST is assigned lower likelihood, as expected. However, the model trained on FashionMNIST predicts higher likelihoods for MNIST images. The majority ($\approx 75\%$) of the MNIST data points get mapped into the mode of the Fashion-MNIST model corresponding to sandals, which is the class with the largest fraction of pixels that are zero. Similarly, for the model trained on MNIST the image of all zeros has very high likelihood and gets mapped to the mode corresponding to the digit $1$ which has the largest fraction of empty space.

## G    EXPECTED DISTANCES BETWEEN GAUSSIAN SAMPLES

Consider two Gaussians with means sampled independently from the standard normal $\mu_1, \mu_2 \sim \mathcal{N}(0, I)$ in $D$-dimensional space. If $s_1 \sim \mathcal{N}(\mu_1, I)$ is a sample from the first Gaussian, then its expected squared distances to both mixture means are:

$$\mathbb{E}\left[\|s_1 - \mu_1\|^2\right] = \mathbb{E}\left[\mathbb{E}\left[\|s_1 - \mu_1\|^2 | \mu_1\right]\right] = \mathbb{E}\left[\sum_{i=1}^{D} \mathbb{E}\left[(s_{1,i} - \mu_{1,i})^2 | \mu_{1,i}\right]\right]$$

$$= \mathbb{E}\left[\sum_{i=1}^{D} \left(\mathbb{E}[s_{1,i}^2] - 2\mu_{1,i}^2 + \mu_{1,i}^2\right)\right] = \mathbb{E}\left[\sum_{i=1}^{D} \left(1 + \mu_{1,i}^2 - \mu_{1,i}^2\right)\right] = D$$

$$\mathbb{E}\left[\|s_1 - \mu_2\|^2\right] = \mathbb{E}\left[\mathbb{E}\left[\|s_1 - \mu_2\|^2 | \mu_1, \mu_2\right]\right] = \mathbb{E}\left[\sum_{i=1}^{D} \mathbb{E}\left[(s_{1,i} - \mu_{2,i})^2 | \mu_{1,i}, \mu_{2,i}\right]\right]$$

$$= \mathbb{E}\left[\sum_{i=1}^{D}\left(1 + \mu_{1,i}^2 - 2\mu_{1,i}\mu_{2,i} + \mu_{2,i}^2\right)\right] = 3D$$

For high-dimensional Gaussians the random variables $\|s_1 - \mu_1\|^2$ and $\|s_1 - \mu_2\|^2$ will be concentrated around their expectations. Since the function $\exp(-x)$ decreases rapidly to zero for positive $x$, the probability of $s_1$ belonging to the first Gaussian $\exp(-\|s_1 - \mu_1\|^2)/\left(\exp(-\|s_1 - \mu_1\|^2) + \exp(-\|s_1 - \mu_2\|^2)\right) \approx \exp(-D)/(\exp(-D) + \exp(-3D)) = 1/(1 + \exp(-2D))$ saturates at 1 with the growth of dimensionality $D$.

## H  FLOWGMM AS GENERATIVE MODEL

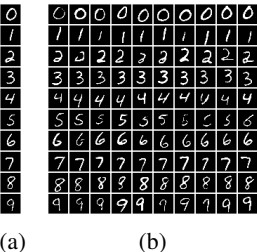

(a)                (b)

Figure 6: Visualizations of the latent space representations learned by supervised FlowGMM on MNIST. **(a)**: Images corresponding to means of the Gaussians corresponding to different classes. **(b)**: Class-conditional samples from the model at a reduced temperature $T = 0.25$.

In Figure 6a we show the images $f^{-1}(\mu_i)$ corresponding to the means of the Gaussians representing each class. We see that the flow correctly learns to map the means to samples from the corresponding classes.Next, in Figure 6b we show class-conditional samples from the model. To produce a sample from class $i$, we first generate $z \sim \mathcal{N}(\mu_i, TI)$, where $T$ is a temperature parameter that controls trade-off between sample quality and diversity; we then compute the samples as $f^{-1}(z)$. We set $T = 0.25^2$ to produce samples in Figure 6b. As we can see, FlowGMM can produce reasonable class-conditional samples simultaneously with achieving a high classification accuracy (99.63%) on the MNIST dataset.

