# OpenReview forum: "Semi-Supervised Learning with Normalizing Flows"
_ICLR.cc/2020/Conference — Reject_

### Official Review · AnonReviewer1 · 2019-10-20
**Official Blind Review #1**

**Rating:** 6

**Review:**

The paper describes how to use normalising flows for Semi Supervised Learning (SSL). Briefly, the method consists in finding a (bijective) map for transforming a mixture of Gaussians into a density approximating the empirical data-distribution -- as usual for flow methods, the parameters are found through likelihood maximisation. This is a elegant approach that naturally exploits the standard so-called cluster assumption in SSL. The papers also shows how to incorporate a consistency-based regularisation within the method.


Although it is an elegant and simple approach, and the article is relatively well written, I think that the paper should be rejected because (1) on image classification tasks (and even with consistency regularisation), the performances are well-below the straightforward-to-implement \Pi-model. (2) for tabular/NLP data, although the performances seems to be good, the comparison with standard methods could have been much better done -- I am still not convinced by the method.

I agree with the authors that there are many situations where it is not possible to find good perturbation (eg. NLP / tabular / genomics / etc...). If the authors could demonstrate more carefully that their approach does lead to state-of-the-art performances in this type of situations, I do believe that the approach would be of great interest. Given that the methods does not work well at all for image classification, I think that he authors should have been much more careful with the comparisons with the standard methods when investigating the performances on NLP/tabular tasks.


(1) basic k-NN benchmark?
(2) basic dimension reduction (PCA / autoencoder / extract lower representation from a NN) associated with either k-NN or label-propagation?
(3) it is *not* difficult at all to implement label propagation with fast nearest-neighbours (eg. FAISS library) and sparse linear algebra on the full datasets. In the current submission, it has not been done for the NLP datasets.
(4) There are indeed several ways to compute distance / affinity within label-propagation-type approaches
(5) Brief description of parameter tuning for label-prop should be added

I think that the method has a lot of potential and the fact that it is not competitive for computer vision task is not important. I encourage the authors to carry out more convincing numerical comparisons ing tabular/NLP/etc.. settings in order to strengthen the message of the paper. If convincing results can be obtained, I believe that the method has a lot of potential.

[Edit after rebuttal]
I would like to thank the authors to have provided additional label propagation experiments and details -- the proposed method appears to be quite much better than this baseline approach, which is very reassuring and proves that it is worth exploring further this line of work.

**Experience Assessment:**

I have read many papers in this area.

**Review Assessment: Checking Correctness Of Derivations And Theory:**

I assessed the sensibility of the derivations and theory.

**Review Assessment: Checking Correctness Of Experiments:**

I carefully checked the experiments.

**Review Assessment: Thoroughness In Paper Reading:**

I read the paper thoroughly.

---

> ### Author Response · Authors · 2019-11-15
> **Response to Review #1**
>
> We appreciate the reviewer’s comments and advice. In response to (1), while it’s true that our model does not perform as well as the Pi model (Tarvainen and Valpola, 2017), the base network architecture in that work is substantially more powerful owing to it not being constrained with invertibility. When trained using all of the labels on CIFAR10 and no unlabeled data, the CNN from Tarvainen and Valpola (2017) has an error rate of 5.56 and the RealNVP architecture we use gets an error rate of 11.55.
>
> The second point made was that although FlowGMM performance on NLP/tabular tasks is promising, the experiments needed a more thorough and careful comparison to supervised and semi-supervised baselines. We agree and have added additional baselines to this section. The performance of k-NN is very similar to the other supervised only methods on the UCI datasets but on the two text classification datasets the performance is substantially worse. We suspect this has to do with the way the BERT embeddings were originally trained for separation type tasks. The label propagation baseline applied in the paper uses a dense affinity matrix, hence challenges with scaling but we thank the reviewer for their suggestion and updated the semi-supervised baselines to include sparse k-NN based label spreading approach that uses a larger fraction of unlabeled data. For tuning the hyper parameters of these label spreading methods, we perform an independent grid search for each method on each dataset.

---

### Official Review · AnonReviewer2 · 2019-10-23
**Official Blind Review #2**

**Rating:** 1

**Review:**

The authors propose a semi-supervised learning model, named as Flow Gaussian Mixture Model (FlowGMM). The model is learnt by maximizing the join likelihood of the labeled and unlabeled data with a consistency regularization.
The authors demonstrate that the proposed model outperforms others on text classification; for image classification, the performance can be improved in future.
Also the authors demonstrate that the model is interpretable via feature visualizations.
Overall the paper is fine written.
Yet, The conclusion is not fairly supported and the paper could be much stronger with the issues discussed already but I don’t think its current form is ready yet.

Below are more detailed comments:
0) It would be nice to add the definition of the performance metric; without the definitions, none of the numbers in the tables would make sense.
1) The main result for text classification in Table 1 is reporting the best of 3 runs, which can’t support the conclusion that the proposed method outperforms the other. In general, it’s nice to provide statistical significance comparing two models or reporting the mean and std across multiple runs.
2) In Table 2, it’s not clear what conclusion could be drawn by comparing the performance of supervised and semi-supervised performance. Are the testing data points the same?
3) The feature visualization as discussed in Section 6.3 is not explained clear. Specifically, “giving us insight into the workings of the model” is not clear; what exactly insight can we get and what exactly are the workings can we get?


**Experience Assessment:**

I have read many papers in this area.

**Review Assessment: Checking Correctness Of Derivations And Theory:**

I assessed the sensibility of the derivations and theory.

**Review Assessment: Checking Correctness Of Experiments:**

I assessed the sensibility of the experiments.

**Review Assessment: Thoroughness In Paper Reading:**

I read the paper at least twice and used my best judgement in assessing the paper.

---

> ### Author Response · Authors · 2019-11-15
> **Response to Review #2**
>
> 0) We updated the caption of the Tables and specified the performance metrics.
> 1) We found that the performance of the methods in Table 1 had high variance, so we decided to adopt the following strategy. We train each method three times, and pick up the run that attained the best accuracy on a validation set. We then report the performance of that run on the test data (different from validation). This procedure is still fair, and is attainable in practice. In an updated version of the paper we will report the mean and std over multiple repetitions of this process.
>
> 2) The performance of a supervised model (which was trained with all labels) shows the general capacity of the model. For example, on CIFAR-10, FlowGMM Sup (All labels), 2nd row of Table 2, is trained on 50k labeled examples, while FlowGMM Sup ($n_l$ labels), 5th row of Table 2, is trained on 4k labeled examples (unlabeled data is not used); reporting both accuracies shows the gap which appears when using much less data, and this gap is significantly decreased when we add unlabeled data. The testing data for a fixed dataset is the standard test split, and is the same across all models and all settings (supervised and semi-supervised).
>
> 3) Like other methods of feature visualization (regularized optimization and inversion by optimization) our novel feature visualization method gives insight into what kinds of features activate a given channel and spatial location, a tool for understanding the intermediate representations and what is learned by the network. Unlike other feature visualization methods, our method does not require optimization or hyperparameters and hence can be performed at real time rates for interactive feature exploration.

---

### Official Review · AnonReviewer3 · 2019-10-24
**Official Blind Review #3**

**Rating:** 1

**Review:**

The paper describes a normalising flow with the prior distribution represented by a Gaussian mixture model (GMM). The method, FlowGMM, maps each class of the dataset to a Gaussian distribution in the latent space by optimising the joint likelihood of both labelled and unlabelled data, thus making the method useful for semi-supervised problems. Predictions are made using the maximum a posteriori estimate of the class label. To make the method robust to small perturbations of the inputs, the authors introduce a novel consistency regularisation term to the total loss function, which maximises the likelihood of predicting the same class after a perturbation.
The authors further examine the learnt latent space by considering two simple, synthetic datasets that can be easily visualised, showing that the latent space behaves in a way one would intuitively expect.
The method is evaluated on both tabular and image data, showing promising results in terms of accuracy (presumably, see below). As the model is found to be overconfident in its predictions, the authors introduce a calibration scheme and empirically verify that it improves the uncertainty estimates. Lastly, the authors introduce a feature visualisation scheme and use it to illustrate the effect of perturbing the activations of the invertible transformations.

I generally like the proposed method, which seems useful and intuitive. I am particularly happy with the discussion on uncertainty calibration, where the authors suggest an elegant addition to the model to increase the variance of the mixture components. I do, however, have significant concerns about the novelty of the paper as well as its structure and clarity, as detailed below. I do, therefore, not recommend it for acceptance.

The paper reads well, although I feel that it lacks some details and explanations. For example, in table 2, it is never mentioned what "FlowGMM Sup" refers to and if it is different from "FlowGMM Supervised". It is also not clear what "(All labels)" refers to - does it mean that labels were provided for the entire dataset or that the models were trained only on the small subset with labels? Or something else? Which performance metric is used in the tables? The accuracy, presumably, but this is never specifically stated. Similarly, the number of datapoints and ratio of labelled to unlabelled data for the synthetic datasets are not reported. They are not crucial to know but should be included for completeness.

While the first half of the paper is informative and well-structured, the second half appears a bit less so. From experimentally verifying that the method works, the paper goes on to discuss uncertainty calibration, examine the latent space representations, and visualising the effect of feature perturbations. While I greatly appreciate the focus on interpreting the trained model, I think it appears somewhat chaotic, as if the authors tried to squeeze too much into the paper. For example, the feature visualisation technique is quite neat, but it works for any flow and is not really used for anything in the paper. I would suggest saving it for a dedicated paper.

I am not convinced by the novelty of this paper. The authors list two contributions: 1) the model itself, 2) an empirical analysis with much focus on the interpretability of the model. While the model is, to my knowledge, indeed novel, the analysis is quite standard, and the interpretability even appears to be oversold. GMMs are nice and intuitive, but not novel in any way, yet the authors seem to be describing the properties of GMMs as specific to their method.
In particular, the authors go to great lengths to show that the latent space representations cluster around the means of the mixture components and that the decision boundary lies in low-density regions of the latent space. I do not see why these properties should be so surprising since the method directly optimises the likelihood of the data under the mixture distribution. That this is also empirically observed is, of course, reassuring, but these observations are better suited for the appendix, in particular given that the paper went over the recommended page limit.
I think that much of the claimed second contribution follows directly from the GMM aspect of the model. Instead of claiming the standard GMM properties as contributions, I think the proposed consistency loss term should be highlighted as a contribution on its own. I find it elegant, and I guess it would be particularly useful for NLP tasks where sentences can be phrased in different ways but still mean the same.

Instead of discussing the latent space, I would have preferred to see extra evaluations of the method, like convergence rates of both FlowGMM and FlowGMM-cons compared to the competing models. Furthermore, a major limitation of the model is that knowledge of the correct number of classes in the data - even in the unsupervised setting. The authors hint at extensions to mitigate this in the discussion (using a Chinese Restaurant Process GMM or by adding extra Gaussians to the mixture during training), but these should have been investigated in the current paper.

In conclusion, I think that the paper lacks novelty and that it spends far too much space on "trivial" properties of the model instead of addressing shortcomings, like the prior specification of the number of classes, which the authors even point out in the discussion.

Minor comments:
- p 4, bottom: "each with 1 hidden layers" -> "each with 1 hidden layer"
- p 5, middle: "FlowGMM is able to leveraged" -> "FlowGMM is able to leverage"
- p 5, bottom: "Table 5.1" -> "Table 1"



**Experience Assessment:**

I have read many papers in this area.

**Review Assessment: Checking Correctness Of Derivations And Theory:**

I assessed the sensibility of the derivations and theory.

**Review Assessment: Checking Correctness Of Experiments:**

I assessed the sensibility of the experiments.

**Review Assessment: Thoroughness In Paper Reading:**

I read the paper at least twice and used my best judgement in assessing the paper.

---

> ### Author Response · Authors · 2019-11-15
> **Response to Reviewer #3**
>
> We thank the reviewer for detailed comments. We addressed the clarity issues that the reviewer identified in the updated version of the paper. In particular,
> FlowGMM Sup and FlowGMM Supervised were both referring to the same method (we renamed the entries in the updates version), FlowGMM trained only using the labeled data. In Table 2, “FlowGMM Sup (All labels)” was trained in a fully-supervised setting, when labels are available for all data points (e.g. 50k labels in CIFAR-10), and “FlowGMM Sup ($n_l$ labels)” was trained only on $n_l$ labeled data (e.g. 4k data points for CIFAR-10).
> In all Tables we use classification accuracy as the predictive metric.
>
> Regarding the novelty of our model interpretation experiments, while we agree that GMMs are not novel, we believe that the combination of normalizing flows with GMMs is novel. We argue that while we could expect some of the observed properties would hold, they are not trivial and verifying them is important. In particular,
> If the data was generated from the FlowGMM model, we would indeed be sure a priori that the decision boundary between classes was passing through a low-density region. However, when we fit actual image data using FlowGMM the fit is not perfect, and there is no way of concluding that the same property would hold without experimentally verifying it. Further, the separation between classes in the latent space is of crucial importance for interpretation of FlowGMM, so we believe that it is important to study it explicitly.
> Another important observation about the latent spaces is that including unlabeled data does indeed push the decision boundary away from unlabeled data. This property is desired, and we believe that explicitly demonstrating it helps interpreting FlowGMM.
>
> We agree that using the Chinese Restaurant Process GMM to automatically determine the number of classes in the data is an exciting direction for future work. However, it would require a non-trivial amount of effort, and methodological advancement, and the reviewer pointed out that “the authors tried to squeeze too much into the paper” even with the current content of the paper. We do not agree that not being able to infer the number of classes is a major shortcoming of FlowGMM, as the setting when the number of classes is unknown is not typically considered in semi-supervised literature, and most of the existing semi-supervised methods are also not directly applicable in this setting. We plan to explore inferring the number of unlabeled classes in future work.

---

> > ### Comment · AnonReviewer3 · 2019-11-15
> > **Response to response**
> >
> > Dear authors, thank you for updating the paper and addressing my concerns.
> >
> > I understand your points about examining the latent space and interpretability, but I am still not convinced that the emphasis of the paper is quite right. It is not so much about squeezing even more into the paper, it is more about changing the focus. Though I do appreciate that this is not possible for a rebuttal.

---

### Public Comment · ~Arsenii_Ashukha1 · 2019-10-10
**The relevant paper "Semi-Conditional Normalizing Flows for Semi-Supervised Learning"**

Thank you for the interesting work. I suggest that the paper "Semi-Conditional Normalizing Flows for Semi-Supervised Learning" from ICML Workshop is relevant. The work also uses a class conditional prior in the form of Normalizing flow and GMM. The discussion of the difference between the methods will be useful.

link: https://invertibleworkshop.github.io/accepted_papers/pdfs/INNF_2019_paper_20.pdf

---

### Decision · Program_Chairs · 2019-12-19

**Decision:**

Reject

**Comment:**

This paper offers a novel method for semi-supervised learning using GMMs.  Unfortunately the novelty of the contribution is unclear, and the majority of the reviewers find the paper is not acceptable in present form.  The AC concurs.

---

> ### Author Response · Authors · 2020-03-09
> **Thoughts**
>
> We respectfully disagree with this assessment. The paper makes a variety of contributions, including a method with substantial novelty. Nearly all classifiers are discriminative. Even approaches that use a generator typically involve a discriminator in the pipeline. For example, sometimes one learns a generator on unlabelled data, then recycles the representation as part of a discriminative classifier. Generative models are compelling because we are trying to create an object of interest. The challenge in generative modelling is that standard approaches to density estimation are poor descriptions of high-dimensional natural signals.
>
> The method proposed in this paper, FlowGMM, is arguably one of the only end-to-end fully generative approaches to classification with normalizing flows, which is a very significant point of novelty. Just because the model involves a Gaussian mixture, and Gaussian mixtures have been used in other contexts, does not take away from the novelty. The method also provides a coherent approach to handling both labelled and unlabelled data, which are often treated separately in deep semi-supervised methods. We also propose a new type of probabilistic consistency regularization that significantly improves FlowGMM on image classification problems. And the method is also relatively interpretable and broadly applicable.
>
> We appreciate the feedback and have made several modifications to the paper, including a more visible presentation of the contributions.